# Palm-based diacylglycerol fat dry fractionation: effect of crystallisation temperature, cooling rate and agitation speed on physical and chemical properties of fractions

Razam Ab Latip[1], Yee-Ying Lee[3], Teck-Kim Tang[3], Eng-Tong Phuah[3], Choon-Min Lee[3], Chin-Ping Tan[4] and Oi-Ming Lai[2,3]

[1] Sime Darby Research Sdn Bhd, R&D Research Centre-Downstream, Pulau Carey, Selangor, Malaysia
[2] Department of Bioprocess Technology, Faculty of Biotechnology and Biomolecular Sciences, Universiti Putra Malaysia, Serdang, Selangor, Malaysia
[3] Institute of Bioscience, Universiti Putra Malaysia, Serdang, Selangor, Malaysia
[4] Department of Food Technology, Faculty of Food Science and Technology, Universiti Putra Malaysia, Serdang, Selangor, Malaysia

Corresponding author
Oi-Ming Lai,
omlai@biotech.upm.edu.my

## ABSTRACT

Fractionation which separates the olein (liquid) and stearin (solid) fractions of oil is used to modify the physicochemical properties of fats in order to extend its applications. Studies showed that the properties of fractionated end products can be affected by fractionation processing conditions. In the present study, dry fractionation of palm-based diacylglycerol (PDAG) was performed at different: cooling rates (0.05, 0.5, 1.0, 1.5, 2.0, 2.5 and 3.0°C/min), end-crystallisation temperatures (30, 35, 40, 45 and 50°C) and agitation speeds (30, 50, 70, 90 and 110 rpm) to determine the effect of these parameters on the properties and yield of the solid and liquid portions. To determine the physicochemical properties of olein and stearin fraction: Iodine value (IV), fatty acid composition (FAC), acylglycerol composition, slip melting point (SMP), solid fat content (SFC), thermal behaviour tests were carried out. Fractionation of PDAG fat changes the chemical composition of liquid and solid fractions. In terms of FAC, the major fatty acid in olein and stearin fractions were oleic (C18:1) and palmitic (C16:0) respectively. Acylglycerol composition showed that olein and stearin fractions is concentrated with TAG and DAG respectively. Crystallization temperature, cooling rate and agitation speed does not affect the IV, SFC, melting and cooling properties of the stearin fraction. The stearin fraction was only affected by cooling rate which changes its SMP. On the other hand, olein fraction was affected by crystallization temperature and cooling rate but not agitation speed which caused changes in IV, SMP, SFC, melting and crystallization behavior. Increase in both the crystallization temperature and cooling rate caused a reduction of IV, increment of the SFC, SMP, melting and crystallization behaviour of olein fraction and vice versa. The fractionated stearin part melted above 65°C while the olein melted at 40°C. SMP in olein fraction also reduced to a range of 26 to 44°C while SMP of stearin fractions increased to (60–62°C) compared to PDAG.

## INTRODUCTION

Obesity is a metabolic disease resulting from the increase of body fat. Obesity is referred as a global epidemic due to its rapid growth rate. In 2004, WHO estimates that more than one billion people are overweight and, of these, 300 million can be considered as obese. Obesity happened in both the developed and developing countries. A progressive increase in obesity rate is found in countries such as United States, Brazil, England and Japan (*James et al., 2001*). Prevalence cases of childhood obesity are increasing as well (*Deckelbaum & Williams, 2001*). It is estimated that if no action is taken against this, these figures could double in 2025. Obesity is often correlated with coronary heart disease, diabetes mellitus, and certain types of cancer. This results in an increased in the global anti-obesity market, which is currently about USD 240 billion.

Diacylglycerols (DAG) oil has metabolic characteristics that are distinct from triacylglycerols (TAG) oil. The consumption of DAG oil is claimed to be able to reduce postprandial serum TAG levels and thus beneficial for the prevention and management of obesity (*Lo et al., 2008*). DAG can be produced from various types of vegetable oils including palm oil. However, the slip melting point of palm based DAG fat is high (more than 50°C) hence, limiting its application in food products. Moreover, the high solid fat content of palm-based DAG fat, which is about 17% at body temperature, results in an undesirable mouth feel. Modification such as fractionation is necessary in order to improve their properties.

Fractional crystallization is a reversible modification process, carried out in 2 stages; crystallization and followed by separation (*Kellens et al., 2007*), done through dry-fractionation. Dry fractionation has gained popularity because of its cheaper process and greener technology. Also, there is no harmful effluent, no chemical used and no loss in yield. Because palm oil contains a mixture of high and low melting glycerides, higher melting glycerides will crystallize into solid (stearin) fraction which can be used for margarine and shortening, while the low melting glycerides remain in a liquid form called olein used for frying media.

The dry fractionation process is simply a controlled crystallization of the melted oil, followed by separation of solid from liquid fraction. Three steps are involved in crystallization process; super cooling of the melt, nucleation and crystal growth (*Zaliha et al., 2004*). The separation is an important step since the amount of liquid fat entrapped in the filter cake will affect the physical properties of this fraction, called stearin, to a great extent. The entrapment of the liquid fat is due to occlusion within crystallized particles or aggregates as well as retention between particles (*Hamm, 1995*). The formations of mixed crystals in the form of agglomerated spherulites, which adsorb liquid within crystals, and depend to a considerable extent to the crystallisation conditions employed (*Patience, Hartel & Illingworth, 1999*). The amount of liquid oil remaining between crystals in the filter cake is determined by the number, size, shape and chemical composition of the

**Table 1** Experimental conditions.

| Parameter | Ct | Cr | As |
| --- | --- | --- | --- |
| Ct (°C) | 30, 35, 40, 45, 50 | 35 (constant) | 35 (constant) |
| Cr (°C/min) | 0.05 (constant) | 0.05, 0.5, 1.0, 1.5, 2.0 | 0.05 (constant) |
| As (rpm) | 90 (constant) | 90 (constant) | 30, 50, 70, 90, 110 |

**Notes.**
Ct = crystallisation temperature; Cr = cooling rate; As = agitation speed.

crystals and the mechanism of filtration (*Amer, Kupranycz & Baker, 1985*). In fats and oil industries, dry fractionation is commonly used to produce a value added stearin and oleic fraction such as palm oil, coconut oil, high-olein high-stearic sunflower oil, milk fat (*Bootello et al., 2011*; *Chaleepa, Szepes & Ulrich, 2010*; *Lopez & Ollivon, 2009*). Dry fractionation is preferable compared to the solvent fractionation as no chemical is involved in this process and it is more economical.

In this study, we performed dry fractionation of palm-based DAG (PDAG) fat at various cooling rates (0.05, 0.5, 1.0, 1.5, 2.0, 2.5 and 3.0°C/min), end-crystallisation temperatures (30, 35, 40, 45 and 50°C) and agitation speeds (30, 50, 70, 90 and 110 rpm) to see the effects of these parameters on the properties and yield of the solid and liquid portions of the DAG fat. Fractionation of this functional oil PDAG can help to extend the applications of PDAG in food application with its stearin and olein fraction. Dry fractionation is only applicable for PDAG but not for other soft oils.

# MATERIALS & METHODS

## Materials

PDAG fat was produced from refined palm oil (RBDPO), provided by Sime Darby Jomalina Food Industries Sdn. Bhd. (Telok Panglima Garang, Selangor, Malaysia), through 1,3-lipase glycerolysis using Novozyme 435 lipase (Novozyme, Denmark) according to Malaysia Patent 201004803. Free fatty acid (FFA) and monoacyglycerides (MAG) were removed using short-path distillation (SPD) to achieve DAG purity of more than 80% (w/w). All chemicals used were of analytical grade except for GC and HPLC purposes, the solvents used were HPLC grade.

## Fractional crystallization

The PDAG fat (400 ml) with free fatty acid (FFA) of less than 0.16% was subjected to dry fractionation using Mettler Toledo LabMax (Greifensee, Switzerland) reactor. The oil was first heated in the reactor for 20 min at 70°C with stirring at 100 rpm to destroy all crystals. The oil was then agitated and cooled at controlled manner to the desired end-temperature. The examined process parameters were summarized in Table 1. The oil was hold in the crystalliser for stabilization followed by separation of the semi-slurry into olein and stearin using hydraulic filter press. The slurry was first fed into the filter press with a minimum pressure 2.0 bar/min. The filling period was 10 min with a maximum pressure 6.0 bar/min. The olein and stearin fraction were weighed and analysed.

## Iodine value (IV) determination

IV was determined according to the AOCS official method Cd 1-25 (1993).

## Fatty acid composition (FAC) analysis

Fatty acid composition was determined by the rapid method of AOCS Official Method Cd 14c-94 (1993). Analysis of fatty acid compositions was done by gas chromatography (Model: Autosystem XL, Perkin Elmer, USA). Fatty acids present in oil were first converted to fatty acid methyl esters (FAME) before injecting into polar SP$^{TM}$ (Supelco, Bellefonte, PA) capillary column (0.25 mm i.d. $\times$ 60 m $\times$ 0.2 µm), to obtain the fatty acid profiles. Temperatures maintained in the analysis were column oven: 130°C, injection block: 250°C and detector temperature: 250°C. Carrier gas was nitrogen at 20 psi. The injection volume was 1 µl.

## Analysis of acylglycerol composition

PDAG fat, olein and stearin fractions (100 µl) were dissolved in solutions of acetone: acetonitrile (60:40) (v/v) (900 µl) and then analyzed for triacyglycerol composition using reversed-phased high performance liquid chromatography (Waters 2695, Connecticut, USA) using AOCS method Ce 5C-93. The TG was separated using packed Supercosil$^{TM}$ LC-18 column (25 cm $\times$ 4.6 mm i.d. $\times$ 5 µm) and eluted from the column using an acetone/acetonitrile (25:75 vol/vol) mobile phase at flow rate of 2 ml/min. The sample injection volume was 1 µl. Detection of the TG was done using refractive index detector (Waters 2414, Connecticut, USA).

## Slip melting point (SMP)

SMP was measured according to AOCS Method Cc.3.25 (1993). Capillary tubes were filled with a 1 cm high column of melted fat. The capillary tubes were then rolled against a piece of ice before being chilled in a refrigerator at 101°C for 16 h to solidify the fat. The tubes were subsequently attached with a rubber band to a thermometer and suspended in a 600 ml beaker of boiled distilled water. The bath temperature was adjusted to 8–10°C below the SMP of the sample, and heat was applied using a heating coil element to increase the bath temperature at a rate of 1°C/min. The temperature at which the fat column rises was reported as the SMP.

## Solid fat content (SFC)

SFC was measured according to Malaysian Palm Oil Board (MPOB) Test Method p4.8 (2004) using pulsed nuclear magnetic resonance (NMR) spectrometry (Bruker NMS 120 minispec). The SFC of PDAG fat, olein and stearin fractions was measured at each separation temperature. The sample in the NMR tube was first melted at 70°C for 30 min, followed by chilling at 0°C for 90 min prior to measurement. Melting, chilling and holding of sample were carried out in pre-equilibrated thermostat water bath. The SFC temperature was set to 10, 20, 25, 30, 35, 40, 45, 50, 55, 60, 65°C. The percentage of SFC was based on three measurements.

### Thermal behavior by differential scanning calorimetry (DSC)

Thermal properties of the oil sample (3–5 mg) was measured using a Perkin Elmer DSC Diamond with hyperDSC (PerkinElmer Ins., Bridgeport Avenue, Shelton, USA). The data processor was PerkinElmer Diamond DSC Auto-sampler. Nitrogen (99.99% purity) was use as the purge gas and flowed at 20.0 ml/min. The DSC instrument was calibrated with indium (m.p. 156.6°C).

## STATISTICAL ANALYSIS

Statistical analysis software (Xlstat's, Addinsoft, New York, USA) was used to perform statistical analysis. Analysis of variance (ANOVA) with Duncan's multiple range tests was performed to determine significant of difference at $P < 0.05$. Analysis was conducted in triplicates.

## RESULTS AND DISCUSSION

### IV, SFC and percentage of yield

IV is a measure of the degree of unsaturation of fats and oils. It is one of the parameters commonly used to measure the quality of olein (*Haryati et al., 1998*). Fractionation of PDAG fat changes the chemical composition of liquid and solid fractions. As the crystallisation proceeds, the more unsaturated fatty acid gradually concentrate in liquid phase, known as olein, leaving behind the more saturated in solid phase, stearin. The fatty acid composition is altered, as expected. The unsaturated fatty acid, were present in higher concentrations in olein fractions.

Table 2 shows the effect of different crystallisation temperature on chemical composition of olein and stearin fractions obtained by dry fractionation of PDAG fat. The present result shows that crystallisation temperature has effect on the IV of olein fraction but not on stearin fraction. A clear correlation ($R^2 = 0.812$) between IV of olein and crystallisation temperature was observed. Olein fractionated at higher crystallisation temperature had lower IV. A reduction in IV of olein fraction was observed as the crystallisation temperature increased from 30 to 50°C. However, there is no significant difference ($P > 0.05$) in IV of olein fraction fractionated at crystallisation temperature of 30, 35 and 40°C. As crystallization temperature decreased, more unsaturated components were concentrated at the liquid fraction which contributed to higher IV (Table 2). A clear correlation ($R^2 = 0.9490$) between crystallisation temperature and the amount oleic acid, C18:1, was also observed (Table 2). A significant decrease ($P < 0.05$) in oleic acid resulted in lower amount of monounsaturated fatty acids (MUFA) and polyunsaturated fatty acids (PUFA). One can also observed that saturated fatty acids (SAFA) gradually increased as crystallisation temperature increased and this might be the result from the increasing palmatic acid concentration. This demonstrated that olein fractions have become more saturated thus, contributed to lower IV for olein fractions. This conclusion is in line with the results of *Arnaud & Collignan (2008)* who studied the effect of the temperature and time on crystallisation, filtration and fraction properties on chicken fat fractionation and showed that the longer crystallisation time and the high suspension viscosity might

**Table 2** Effect of different crystallisation temperature on chemical composition of olein and stearin fractions obtained by dry fractionation of PDAG fat.

| Layer | Ct | Acylglycerol Composition (%) | | | Fatty Acid Composition (%) | | | | | IV | Yield (%) |
|---|---|---|---|---|---|---|---|---|---|---|---|
| | | MAG | DAG | TAG | C16:0 | C18:1 | SAFA | MUFA | PUFA | | |
| **Ol** | | **-** | **97.89±0.06** | **2.64±0.07** | **44.08±0.12** | **39.86±0.15** | **49.92±0.15** | **40.01±0.10** | **10.06±0.15** | **49.86±0.25** | **-** |
| | 30 | - | $96.23\pm0.06^c$ | $3.78\pm0.07^b$ | $33.53\pm0.11^e$ | $49.11\pm0.26^a$ | $38.27\pm0.18^e$ | $49.30\pm0.18^a$ | $12.22\pm0.26^a$ | $59.97\pm0.31^a$ | $37.54\pm0.06^e$ |
| | 35 | - | $96.13\pm0.08^c$ | $3.88\pm0.07^b$ | $34.31\pm0.19^d$ | $48.35\pm0.10^b$ | $39.22\pm0.17^d$ | $48.53\pm0.20^b$ | $12.03\pm0.15^a$ | $59.74\pm0.24^a$ | $52.45\pm0.05^c$ |
| | 40 | - | $96.42\pm0.10^b$ | $3.57\pm0.08^c$ | $34.89\pm0.18^c$ | $47.81\pm0.15^c$ | $39.88\pm0.10^c$ | $48.00\pm0.26^c$ | $11.90\pm0.30^a$ | $59.73\pm0.28^a$ | $42.95\pm0.05^d$ |
| | 45 | - | $90.72\pm0.10^d$ | $9.30\pm0.07^a$ | $36.73\pm0.12^b$ | $46.22\pm0.11^d$ | $41.93\pm0.28^b$ | $46.40\pm0.20^d$ | $11.46\pm0.26^b$ | $57.64\pm0.36^b$ | $55.83\pm0.08^b$ |
| | 50 | - | $96.72\pm0.05^a$ | $3.27\pm0.07^d$ | $38.65\pm0.20^a$ | $44.55\pm0.10^e$ | $43.99\pm0.17^a$ | $44.72\pm0.10^e$ | $11.07\pm0.16^b$ | $55.41\pm0.39^c$ | $59.36\pm0.04^a$ |
| St | 30 | - | $99.28\pm0.06^d$ | $0.71\pm0.04^a$ | $67.61\pm0.19^d$ | $16.18\pm0.19^a$ | $75.91\pm0.21^d$ | $19.42\pm0.17^a$ | $4.58\pm0.08^a$ | $24.19\pm0.31^b$ | $62.46\pm0.04^a$ |
| | 35 | - | $99.80\pm0.06^a$ | $0.19\pm0.06^d$ | $67.47\pm0.20^d$ | $19.58\pm0.17^a$ | $75.62\pm0.27^d$ | $19.64\pm0.20^a$ | $4.64\pm0.05^a$ | $24.91\pm0.41^a$ | $47.55\pm0.05^c$ |
| | 40 | - | $99.35\pm0.08^{cd}$ | $0.63\pm0.09^{ab}$ | $70.14\pm0.24^a$ | $17.28\pm0.26^d$ | $78.51\pm0.28^a$ | $17.34\pm0.26^d$ | $4.07\pm0.08^c$ | $21.54\pm0.36^d$ | $57.05\pm0.03^b$ |
| | 45 | - | $99.56\pm0.06^b$ | $0.44\pm0.05^c$ | $69.33\pm0.10^b$ | $17.99\pm0.21^c$ | $77.61\pm0.18^b$ | $18.05\pm0.10^c$ | $4.26\pm0.14^b$ | $22.69\pm0.21^c$ | $44.17\pm0.04^d$ |
| | 50 | - | $99.41\pm0.05^c$ | $0.60\pm0.06^b$ | $68.41\pm0.19^c$ | $18.76\pm0.11^b$ | $76.55\pm0.17^c$ | $18.83\pm0.18^b$ | $4.52\pm0.08^a$ | $23.23\pm0.23^c$ | $40.64\pm0.06^e$ |

**Notes.**

Ct = crystallization temperature = 30, 35, 40, 45, 50°C, Ol = Olein, St = Stearin, SAFA = Saturated fatty acid, MUFA = monounsaturated fatty acid, PUFA = polyunsaturated fatty acid. Each value in table represents the mean ± standard deviation of sample analysis from triplicate analysis. Mean within each column with different superscripts letter $a, b, c, d, e$ are significantly ($P < 0.05$) different, $a, b, c, d, e$ ($P < 0.05$).

possibly preserve the crystalline integrity at low temperatures. In a study by *Mamat et al. (2005)* on palm and sunflower oil blends fractionated using different temperatures, it was reported that higher IV can be obtained due to higher PUFA propotion found in liquid fraction when lower fractionation temperature was applied. However, no correlation ($R^2 = 0.2499$) between IV of stearin and crystallisation temperature was observed in the present study. This is probably related to an inconsistency in separation processes. In our study, separation was done by manual pressing therefore; the pressure and the duration of pressing were not effectively controlled. Hence, an increase in olein entrapment might have contributed to an increase in IV of stearin and vice versa. The IV for PDAG fat (49.86) was intermediate between olein and stearin fractions as PDAG fat has equal proportion of saturated and unsaturated fatty acids (Table 2).

Cooling rate influenced the nature of crystals obtained. The effect of different cooling rates on chemical compositions of olein and stearin fractions obtained by dry fractionation of PDAG fat is shown in Table 3. Similar to the effect of crystallisation temperature, the IV of the olein fraction is influenced by the cooling rate but not the stearin fraction. A clear correlation ($R^2 = 0.7373$) between IV of olein and cooling rate was identified in this study. As cooling rate increased, the iodine value of olein fractions decreased. Table 3 shows SAFA increased while MUFA decreased as cooling rate increased for olein fractions. Significant decrease ($P < 0.05$) in MUFA composition was influenced by the reduction of oleic acid (C18:1). This contributed to lower IV for olein fractions. However there is no significant difference ($P > 0.05$) in IV for cooling rates of 0.05, 0.5, 1.0 and 1.5°C/min. This finding is similar to what was reported by *deMan (1964)* and *Schaap & Rutten (1976)* who found little difference in slip point, solid fat content (SFC), yield, hardness, thermal melting curves, and fatty acid composition over the ranges from 0.01 to 1°C/min of cooling rate. However, in this study, no correlation ($R^2 = 0.010$) between IV and cooling rate was observed for

**Table 3** Effect of different cooling rates on chemical composition of olein and stearin fractions obtained by dry fractionation of palm-based diacylglycerol fat.

| Layer | Cr | Acylglycerol composition (%) | | | Fatty acid composition (%) | | | | | IV | Yield (%) |
|---|---|---|---|---|---|---|---|---|---|---|---|
| | | MAG | DAG | TAG | C16:0 | C18:1 | SAFA | MUFA | PUFA | | |
| **Ol** | | 0.25±0.05 | 86.53±0.10 | 13.23±0.06 | 43.52±0.02 | 39.62±0.07 | 50.49±0.12 | 39.86±0.14 | 9.64±0.22 | 49.51±0.21 | - |
| | 0.05 | 0.30±0.05[b] | 78.04±0.07[b] | 21.66±0.09[c] | 34.98±0.28[c] | 47.10±0.18[b] | 41.27±0.16[d] | 47.29±0.23[b] | 11.44±0.18[a] | 58.83±0.21[a] | 75.56 ± 0.04[b] |
| | 0.5 | 0.40±0.05[a] | 78.06±0.10[b] | 21.54±0.08[c] | 34.44±0.27[d] | 47.53±0.24[a] | 40.68±0.12[c] | 47.73±0.14[a] | 11.61±0.22[a] | 58.44±0.22[a] | 71.68±0.02[d] |
| | 1.00 | 0.25±0.04[b] | 68.00±0.13[d] | 31.75±0.07[a] | 35.27±0.12[bc] | 46.91±0.10[bc] | 41.39±0.15[c] | 47.10±0.13[b] | 11.51±0.22[a] | 58.45±0.35[a] | 76.69±0.03[a] |
| | 1.50 | 0.40±0.06[a] | 77.66±0.07[c] | 21.94±0.06[b] | 35.60±0.17[b] | 46.59±0.21[c] | 41.89±0.27[b] | 46.78±0.15[c] | 11.32±0.23[ab] | 58.38±0.20[a] | 73.49±0.06[c] |
| | 2.0 | 0.25±0.04[b] | 80.34±0.10[a] | 21.94±0.06[a] | 37.19±0.21[a] | 45.28±0.16[d] | 43.52±0.28[a] | 45.46±0.17[d] | 11.01±0.21[b] | 57.32±0.18[b] | 68.85±0.03[e] |
| St | 0.05 | 0.42±0.04[b] | 97.40±0.05[a] | 2.60±0.03[e] | 71.04±0.19[c] | 16.18±0.19[d] | 80.00±0.23[a] | 16.26±0.11[d] | 3.75±0.20[c] | 21.80±0.20[e] | 24.44±0.06[d] |
| | 0.5 | 0.65±0.07[a] | 95.66±0.05[c] | 3.69±0.05[c] | 65.28±0.19[c] | 16.18±0.19[d] | 80.00±0.23[a] | 16.26±0.11[d] | 3.75±0.20[c] | 27.63±0.27[b] | 28.32±0.08[b] |
| | 1.00 | 0.60±0.06[a] | 96.09±0.14[b] | 3.31±0.08[d] | 66.99±0.14[bc] | 19.61±0.24[c] | 75.80±0.11[b] | 19.71±0.16[c] | 4.50±0.26[b] | 24.77±0.15[d] | 23.31±0.09[e] |
| | 1.5 | 0.65±0.07[a] | 94.52±0.08[d] | 4.83±0.05[b] | 65.34±0.16[b] | 20.94±0.28[b] | 74.12±0.16[c] | 21.00±0.23[b] | 4.88±0.10[a] | 27.00±0.30[c] | 26.51±0.09[c] |
| | 2.0 | 0.65±0.07[a] | 94.52±0.08[e] | 4.83±0.05[a] | 63.63±0.28[a] | 22.41±0.22[a] | 72.23±0.10[d] | 22.52±0.13[a] | 5.24±0.16[a] | 28.05±0.20[a] | 31.17±0.03[a] |

**Notes.**

Cr = cooling rate = 0.05, 0.5, 1.0, 2.0 °C/min, Ol = Olein, St = Stearin, SAFA = Saturated fatty acid, MUFA = monounsaturated fatty acid, PUFA = polyunsaturated fatty acid. Each value in table represents the mean ± standard deviation of sample analysis from triplicate analysis. Mean within each column with different superscripts letter $a, b, c, d, e$ are significantly ($P < 0.05$) different, $a, b, c, d, e$ ($P < 0.05$).

the stearin fractions, probably due to inconsistency in the separation steps as mentioned earlier. The increment in IV for the stearin fraction was simply due to the presence of higher quantity of entrained olein in the stearin fractions which resulted in higher stearin yield as showed in Table 3. Increased IV due to olein entrapment was evidenced by the increase and decrease in oleic acid and palmatic acid, respectively, i.e. higher MUFA and lower SAFA contents were detected. According to *deMan (1964)*, a slower crystallisation process will led to a decreased solid fat content, the hardness of milk fat, as well as the aggregation of small crystalline particles into larger crystalline particles.

There are many factors that can influence lipid crystallization. One of the most notable is the process by which the melted sample is cooled down. This includes the cooling rate, crystallisation temperature and also agitation speed. The main function of agitation during fat fractionation was suspending the crystal aggregates and enhancing the heat transfer. Table 4 shows the effect of different agitation speed on chemical composition of olein and stearin fractions obtained by dry fractionation of PDAG fat. For the experimental conditions described here, IV did not seem to be affected by agitation. No correlation between IV and agitation speed was observed in olein ($R^2 = 0.096$) and stearin ($R^2 = 0.139$) fractions. However, stearin fractionated at 50 rpm has highest IV (Table 4) which was possibly because of high olein entrapment resulting in high stearin yield. In contrast with the study conducted by *Vanhoutte et al. (2003)* on the filtration properties and crystallisation kinetics of milk fat fractionation. They performed experiments at 13 to 25 rpm to investigate the effect of higher agitation speed and found that more intense agitation produced softer stearin as a result of more oil inclusion. The result was explained by higher shear rates, which break down crystal aggregates. Agitation rate had a marked effect on crystal size. Higher agitation rate had a dramatic effect on crystal size resulting

**Table 4** Effect of different agitation speed on chemical composition of olein and stearin fractions obtained by dry fractionation of palm-based diacylglycerol fat.

| Layer | As | Acylglycerol composition (%) | | | Fatty Acid composition (%) | | | | | IV | Yield (%) |
|---|---|---|---|---|---|---|---|---|---|---|---|
| | | MAG | DAG | TAG | C16:0 | C18:1 | SAFA | MUFA | PUFA | | |
| Ol | | $0.25\pm0.05$ | $92.53\pm0.10$ | $7.22\pm0.06$ | $43.52\pm0.02$ | $39.62\pm0.07$ | $49.49\pm0.12$ | $40.86\pm0.14$ | $10.64\pm0.22$ | $51.51\pm0.21$ | - |
| | 30 | $0.36\pm0.04^a$ | $93.03\pm0.06^a$ | $6.61\pm0.07^d$ | $33.34\pm0.18^c$ | $49.28\pm0.15^a$ | $38.07\pm0.18^c$ | $49.48\pm0.11^a$ | $12.45\pm0.25^a$ | $61.62\pm0.28^b$ | $38.85\pm0.05^d$ |
| | 50 | $0.38\pm0.04^a$ | $92.39\pm0.04^d$ | $7.23\pm0.05^c$ | $33.85\pm0.18^b$ | $48.80\pm0.20^b$ | $38.64\pm0.28^b$ | $49.00\pm0.30^b$ | $12.38\pm0.12^a$ | $61.18\pm0.30^{bc}$ | $36.12\pm0.10^e$ |
| | 70 | $0.38\pm0.04^a$ | $90.91\pm0.06^e$ | $8.71\pm0.04^a$ | $34.31\pm0.17^a$ | $48.43\pm0.20^b$ | $39.15\pm0.15^a$ | $48.63\pm0.22^b$ | $12.25\pm0.35^a$ | $60.88\pm0.38^c$ | $42.06\pm0.07^c$ |
| | 90 | $0.36\pm0.02^a$ | $92.51\pm0.04^b$ | $7.13\pm0.11^c$ | $34.02\pm0.17^{ab}$ | $48.65\pm0.20^b$ | $38.80\pm0.20^{ab}$ | $48.85\pm0.25^b$ | $12.36\pm0.14^a$ | $61.08\pm0.22^{bc}$ | $50.30\pm0.02^b$ |
| | 110 | $0.33\pm0.04^a$ | $91.96\pm0.09^d$ | $7.71\pm0.05^b$ | $33.06\pm0.16^c$ | $49.47\pm0.25^a$ | $37.77\pm0.20^c$ | $49.67\pm0.27^a$ | $12.56\pm0.26^a$ | $62.18\pm0.27^a$ | $54.92\pm0.08^a$ |
| St | 30 | $0.25\pm0.04^{ab}$ | $95.36\pm0.06^c$ | $4.39\pm0.07^a$ | $68.32\pm0.16^a$ | $18.71\pm0.19^d$ | $76.75\pm0.18^a$ | $18.78\pm0.23^d$ | $4.48\pm0.25^b$ | $23.65\pm0.25^d$ | $61.15\pm0.06^b$ |
| | 50 | $0.27\pm0.03^{ab}$ | $95.91\pm0.06^b$ | $3.82\pm0.04^b$ | $64.54\pm0.37^d$ | $21.99\pm0.21^a$ | $72.58\pm0.28^d$ | $22.08\pm0.22^a$ | $5.36\pm0.15^a$ | $27.68\pm0.19^a$ | $63.88\pm0.03^a$ |
| | 70 | $0.23\pm0.04^b$ | $97.68\pm0.06^a$ | $2.09\pm0.07^c$ | $66.14\pm0.16^b$ | $20.55\pm0.25^c$ | $74.37\pm0.23^b$ | $20.63\pm0.37^c$ | $4.98\pm0.26^a$ | $25.62\pm0.18^c$ | $57.95\pm0.05^c$ |
| | 90 | $0.31\pm0.04^a$ | $95.33\pm0.04^c$ | $4.36\pm0.05^a$ | $65.80\pm0.17^b$ | $20.86\pm0.29^c$ | $74.01\pm0.21^b$ | $20.94\pm0.26^c$ | $5.05\pm0.38^a$ | $25.71\pm0.24^c$ | $49.71\pm0.09^d$ |
| | 110 | $0.24\pm0.03^b$ | $95.91\pm0.03^b$ | $3.85\pm0.06^b$ | $65.17\pm0.33^c$ | $21.42\pm0.18^b$ | $73.29\pm0.29^c$ | $21.49\pm0.21^b$ | $5.22\pm0.18^a$ | $26.36\pm0.12^b$ | $45.08\pm0.02^e$ |

**Notes.**

As = agitation speed = 30, 50, 70, 90, 110 rpm, Ol = Olein, St = Stearin, SAFA = Saturated fatty acid, MUFA = monounsaturated fatty acid, PUFA = polyunsaturated fatty acid. Each value in table represents the mean ± standard deviation of sample analysis from triplicate analysis. Mean within each column with different superscripts letter $a, b, c, d, e$ are significantly ($P < 0.05$) different, $a, b, c, d, e$ ($P < 0.05$).

in formation of many small crystals (*Herrera & Hartel, 2000*), which is perhaps evidence of secondary nucleation caused by crystal contact mechanism (*Hartel, 2001*). *Martini, Herrera & Hartel (2002)* reported that blends of a high milk fat fraction and sunflower oil crystallised without agitation appeared to be more densely arranged within the crystal and to have bigger crystal sizes than samples crystallized in dynamic conditions. *Breitschuh & Windhab (1996)* showed that higher agitation promotes co-crystallization, probably due to an enhanced heat transfer.

The stearin yield is strongly dependent on the crystallisation temperatures and agitation speed but not cooling rate (Tables 2, 3 and 4). As crystallisation temperature increased, the yield of stearin fractions decreased (Table 2). This is because fewer crystals were formed at higher temperature hence reducing the amount of solid fractions. At the same time, intense agitation resulted in formation of small crystals which reduced the amount of solid fraction. *Herrera & Hartel (2000)* found that higher agitation rates led to formation of smaller fat crystals in a milkfat model system. The formation of smaller crystal resulted in poor separation hence reduced the amount of solid fraction. Similar result was also reported by *Vanhoutte et al. (2003)* on the effect of crystallisation temperatures but not agitation speed.

The acylglycerol composition can be altered from the feed oil as expected. The propotion of TAG was higher in the olein, whereas DAG is concentrated in the stearin fraction (Tables 2, 3 and 4). The influence of process parameters on the glyceride composition was insignificant compared to the changes in physical properties hence it was not investigated.

## SFC

Figure 1A shows the SFC of PDAG fat and its fractions at different cooling rates. The olein fractions curves showed lower SFC compared to PDAG fat. It was observed that olein fractionated at lower cooling rates of 0.05, 0.5 and 1.0°C/min were completely melted at lower temperature (40°C) whereas, the SFC for olein fractionated at 1.5 and 2.0°C/min still retained at 0.97% and 3.1%, respectively. At cooling rate of 2.0°C/min, the SFC of olein fraction was highest due to high content of SAFA and low amount of unsaturated fatty acids. As crystallisation temperature decreased, the SFC for the olein fractions decreased. Ol 30 and Ol 35 completely melted at body temperatures whereas, the solid fat of Ol 40, Ol 45 and Ol 50 still retain at 2.42, 2.82 and 8.32%, respectively (Fig. 1B). Agitation speed does not affect the SFC of the olein fractions (Fig. 1C). The SFC of stearin fractions (Figs. 1A–1C) are all higher than the SFC of palm-based DAG fat and were melted at temperatures above 65°C. This is due to the fact that the stearin fractions contained higher amounts of saturated fat with high melting points and were crystallized out at higher temperatures during the fractionation process. One can observed that the SFC of stearin fractions is not influenced by these three parameters.

## SMP

The SMP of PDAG, olein and stearin fractionated at various cooling rates, crystallisation temperature and agitation speeds are shown in Table 5. PDAG fat has a SMP of 51.6°C. Upon fractionation, the SMP for the olein fractions were reduced to a range of 26 to 44°C. In contrast, the SMP of stearin fractions increased (60–62°C). This is due to an increase in the amount of saturated fatty acid (C16:0) and decreased in unsaturated fatty acids (C18:1) in the stearin fractions, as indicated in Table 2 to Table 4. SMP of the olein but not stearin fraction is influenced by the cooling rate. No significant difference ($P > 0.05$) in the SMP of the olein fractions was observed as the cooling rate increased from 0.05 to 1.00°C/min. However, fractionation of PDAG fat at higher cooling rate showed significant increase ($P < 0.05$) in the SMP of the olein fractions. Large increase in SMP was observed when cooling rate increased from 1.50 to 2.00°C/min. Based on our DSC study (Fig. 2C), olein fractionated at 2.00°C/min contained high proportion of high melting components resulting in high SMP. However, no significant difference ($P > 0.05$) was observed in the SMP of stearin fractions. Fractionation at higher crystallization temperature showed significant increased ($P < 0.05$) in SMP of olein fractions (Table 5).

SMP of the olein fraction is affected by the crystallization temperature, but not for the stearin fraction. The SMP of olein increased from 28.6 to 44°C as crystallisation temperatures increased from 30 to 50°C. The melting thermogram (Fig. 2G) showed an increase in high melting components as crystallisation temperature increased. Increasing the crystallization temperature leads to the broadening of the crystallization exotherm which refers to a longer crystallisation process (Fig. 2E). This shows that olein fractions have become more saturated thus, contributed to higher SMP for olein fractions. Insignificant differences in the SMP of stearin fractions indicated that crystallisation temperature does not influence the SMP of the stearin. An increase in agitation speed

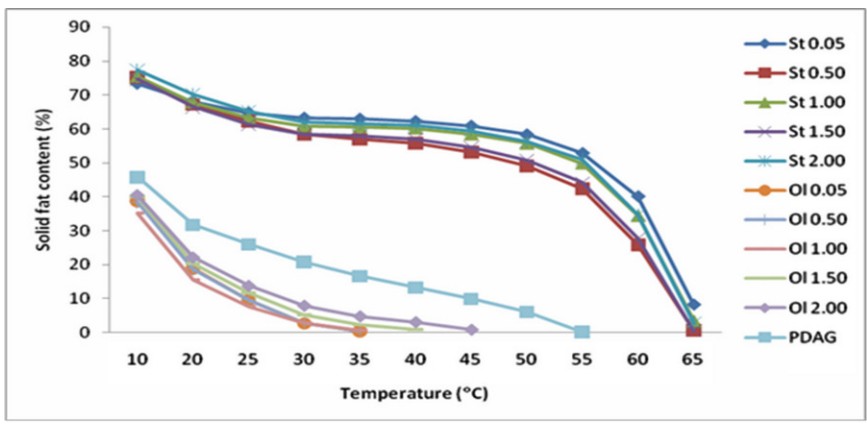

1A

1B

1C

**Figure 1** (A) Solid fat content of PDAG fat, stearin and olein fractionated at different cooling rate. St, stearin; Ol, olein; cooling rates in °C/min, 0.05, 0.50, 1.00, 1.50 and 2.00. (B) Solid fat content of palm-based DAG fat, stearin and olein fractionated at 

**Figure 1 (...continued)**
different crystallization temperature. St, stearin; Ol, olein; crystallization temperature in °C, 30, 35, 40, 45 and 50. (C) Solid fat content of palm-based DAG fat, stearin and olein fractionated at different agitation speed. St, stearin; Ol, olein; agitation speed in rpm, 30, 50, 70, 90 and 110.

**Table 5** Slip melting point of PDAG olein and stearin.

| | Cooling rate (°C/min) | | | | |
|---|---|---|---|---|---|
| | 0.05 | 0.50 | 1.00 | 1.50 | 2.00 |
| **Olein** | $30.0\pm0.50^c$ | $30.0\pm0.50^c$ | $30.0\pm0.20^c$ | $31.8\pm0.20^b$ | $37.8\pm0.20^a$ |
| **Stearin** | $62.5\pm0.20^a$ | $61.6\pm0.30^b$ | $61.8\pm0.30^b$ | $62.0\pm0.30^b$ | $62.0\pm0.10^b$ |
| | Crystallization temperature (°C) | | | | |
| | 30 | 35 | 40 | 45 | 50 |
| **Olein** | $28.6\pm0.20^e$ | $28.0\pm0.20^d$ | $32.0\pm0.20^c$ | $34.4\pm0.20^b$ | $44.0\pm0.30^a$ |
| **Stearin** | $62.0\pm0.20^b$ | $62.0\pm0.40^b$ | $62.4\pm0.20^{ab}$ | $62.6\pm0.26^a$ | $62.6\pm0.20^a$ |
| | Agitation speed (rpm) | | | | |
| | 30 | 50 | 70 | 90 | 110 |
| **Olein** | $26.5\pm0.26^c$ | $27.0\pm0.20^b$ | $27.4\pm0.20^{ab}$ | $27.1\pm0.26^{ab}$ | $27.5\pm0.20^a$ |
| **Stearin** | $61.4\pm0.17^a$ | $60.2\pm0.20^b$ | $60.2\pm0.36^b$ | $60.2\pm0.20^b$ | $60.2\pm0.10^b$ |

**Notes.**
Each value in table represents the mean ± standard deviation of sample analysis from triplicate analysis. Mean within each column with different superscripts letter $a, b, c, d, e$ are significantly ($P < 0.05$) different, $a, b, c, d, e$ ($P < 0.05$).

showed no significant difference in the SMP of both olein and stearin fractions indicating that fractionation of PDAG fat is not influenced by agitation speed.

## Thermal behavior properties

Figure 2 shows the crystallisation and melting curves of PDAG fat and its fractions at different cooling rate, crystallisation temperature and agitation speeds. The crystallisation and melting curves recorded by DSC showed different crystallisation and melting peaks for PDAG and its fractions. Two major endothermic peaks; 53.78 and 23.41°C and one minor peak; −4.34°C were observed in PDAG fat melting thermogram (Fig. 2C). PDAG stearin though showed two melting peaks and like its parent fats, the proportion of these peaks are different and also the first peak (Pk1) is shifted towards a higher temperature (Figs. 2D, 2H and 2L).

The proportion of low melting peak is reduced and that of the higher peaks are increased in stearin compared to its original fat due to the removal of the liquid fraction, which is reflected in SFC (Figs. 1A to 1C) and melting profiles (Figs. 2A to 2L). The SFC at all temperatures is increased in stearin compared to the original fat and thus the plasticity is improved (Figs. 2A to 2L). PDAG stearin showed one exothermic peak which attributed to the high melting component being shifted to a higher temperature compared to its original fat and this is expected due to the removal of liquid fraction (Figs. 2B, 2F and 2J). The stearin fractions did not show differences in melting and crystallization behaviors with changes in processing parameters. One can conclude that the melting properties of stearin fractions were not influenced by the cooling rate, crystallisation temperatures and agitation speeds.

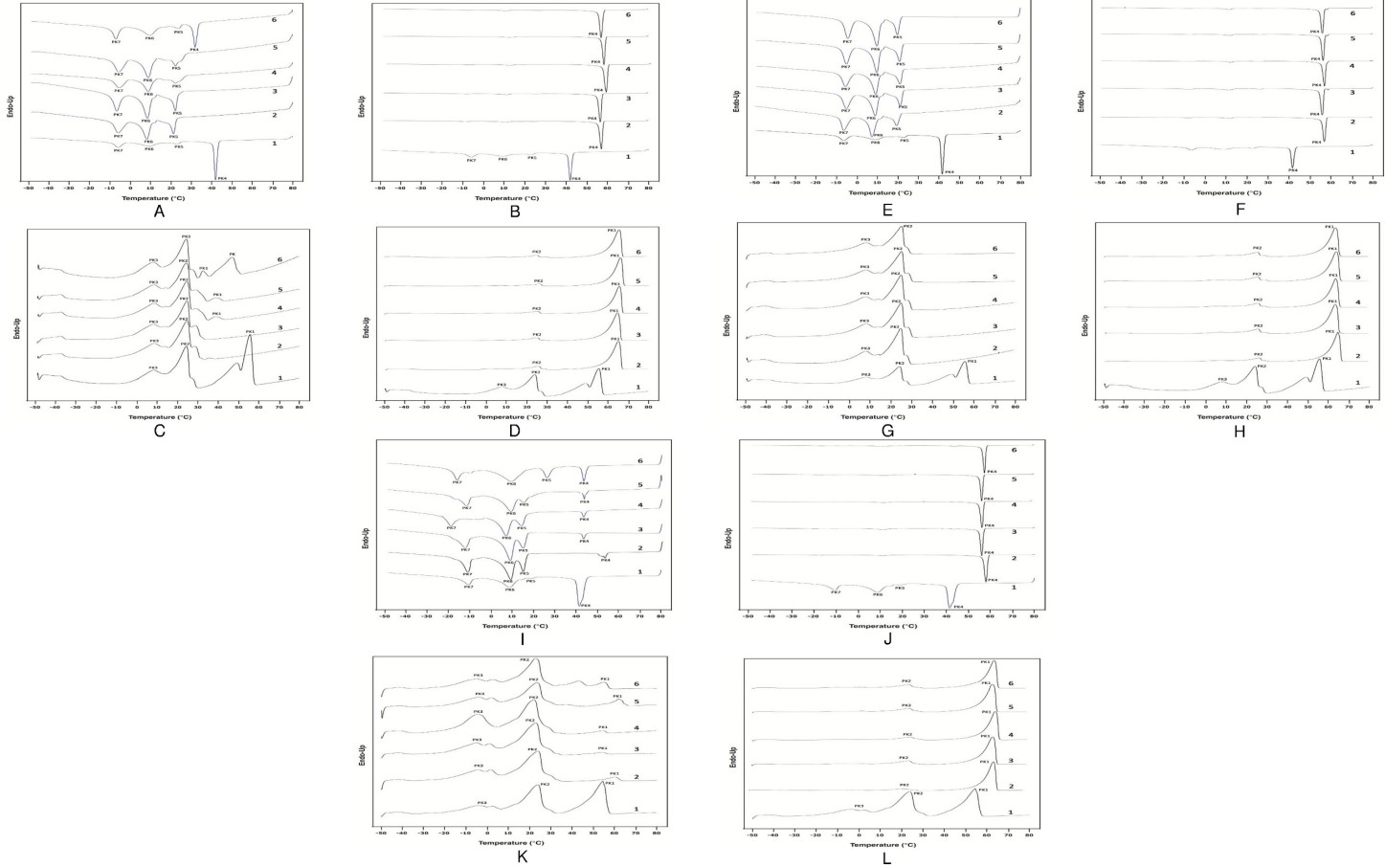

**Figure 2** DSC crystallization (A, B, E, F, I, J) and melting curve (C, D, G, H, K, L) for olein (A, C, E, G, I, K) and stearin fraction (B, D, F, H, J, K) at different cooling rate (A–D), crystallization temperature (E–H) and agitation speed (I–L). (A–D) 1; PDAG fat, Olein fractionated at 2; 0.05°C/min, 3; 0.50°C/min, 4; 1.00°C/min, 5; 1.50°C/min and 6; 2.00°C/min. (E–H) 1; PDAG fat, Olein/stearin fractionated at 2; 30°C, 3; 35°C, 4; 40°C, 5; 45°C and 6; 50°C. (I–L) 1; PDAG fat, Olein/stearin fractionated at 2; 30 rpm, 3; 50 rpm, 4; 70 rpm, 5; 90 rpm and 6; 110.

The olein fractionated at different cooling rates showed 4 exothermic peaks similar to PDAG fat, indicating that the high melting components were not completely removed (Fig. 2A). However, olein showed very small high melting peak unlike parent fat or stearin part. The olein though showed 3 melting peaks like PDAG fat, but the proportion of these peaks are different (Fig. 2C). The proportion of high melting peak is reduced. One can conclude that fractionation of PDAG fat at cooling rates from 0.05 to 2.00°C/min produced harder olein. This is supported by the SFC profile which showed complete melting at temperature above 35°C (Fig. 1A).

Fractionation at higher crystallisation temperature produced olein with high proportion of high melting components. Cooling and melting thermograms (Figs. 2E and 2G) of olein fractionated at 50°C displayed one peak at higher temperature region at 32.68 and 45.99°C, respectively and this was attributed to the high melting components, which contributed to higher SFC (Fig. 1B). However, olein obtained from removal of stearin at lower temperatures showed two melting peaks and three exothermic peaks

at lower temperature regions due to the removal of higher melting fractions. The olein fractionated at different agitation speed showed three crystallisation peaks (Fig. 2I) and two endothermic peaks (Fig. 2K). One can observe that agitation speed does not influence the melting and cooling properties of the olein fractions.

## CONCLUSION

The dry fractionation process is environmentally friendly and a well established method used in the fats and oils industry to separate the olein (liquid fraction) and stearin fractions (solid fraction) of fats and oils so as to extend their applications especially in the food industry. It is important to know that different types of fats and oils require different types of operating parameters during fractionation. The influence of these parameters such as crystallization temperature, cooling rate and agitation speed during the fractionation process need to be taken into consideration in order to obtain a good quality olein and stearin fraction. In the present study, the effect of crystallisation temperature, cooling rate and agitation speed on physical (DSC, SMP, thermal behaviour) and chemical properties (IV) of the stearin and olein of palm-based diacylglycerol was studied. Crystallisation temperature and cooling has an influence on the chemical property of the olein fraction but not for the stearin fraction. On the other hand, agitation speed does not have any affect on both the chemical properties of olein and stearin fraction.

### Funding
The authors would like to thank Sime Darby Research Sdn Bhd for their financial support. The funders had no role in study design, data collection and analysis, decision to publish, or preparation of the manuscript.

### Grant Disclosures
The following grant information was disclosed by the authors:
Techno Fund from Ministry of Science, Technology and Innovation: 6370100.

### Competing Interests
Prof. Dr. Lai Oi Ming is an Academic Editor for PeerJ.

### Author Contributions
- Razam Ab Latip and Oi-Ming Lai conceived and designed the experiments, performed the experiments, analyzed the data, contributed reagents/materials/analysis tools, wrote the paper.
- Yee-Ying Lee and Choon-Min Lee contributed reagents/materials/analysis tools, wrote the paper.

- Teck-Kim Tang and Eng-Tong Phuah performed the experiments, contributed reagents/materials/analysis tools, wrote the paper.
- Chin-Ping Tan designed the experiments.

## Patent Disclosures

The following patent dependencies were disclosed by the authors:

Malaysian Patent Applications No: PS 201004803, Filing Date: 12/10/10.

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
