# Peer review of "Palm-based diacylglycerol fat dry fractionation: effect of crystallisation temperature, cooling rate and agitation speed on physical and chemical properties of fractions"

_PeerJ, doi:10.7717/peerj.72_

## Round 0.1 · original submission · Major Revisions

Data provided in this paper demonstrated the feasiblity of separating stearin and olein of the Palm-based Diacylglycerol fat by dry fractionation. I hope the authors may wish to restructure the paper in a more logical way as Reviewer 2 suggested.

Reviewer 1 ·

Basic reporting

This is really interesting work since diacylglycerols now are used as functional oils. This authors try to separate stearin and olein of the Palm-based Diacylglycerol fat by dry fractionation. Parameters like cooling rates , end-crystallisation temperatures and agitation speeds have been investigated. Physicochemical properties of olein and stearin fraction have been detected. The manuscript is well written in English. The data is adequate and the issues have been well addressed. The results will be helpful to the readers in this field. I recommend the publication of this work in the journal.

Experimental design

the experiments are well designed and statistical analysis has been used for the data.

Validity of the findings

the findings have been well addressed according to the data. However, the resolution of the figures especially, the DSC curves, is not enough.

Additional comments

The authors applied a common technology, dry fractionation, into an intersting stuff, the palm based diacylglycerols. The findings of the work is useful for potential application of this functional fat into more food products.

Reviewer 2 ·

Basic reporting

Introduction part:
It explains the situation and the importance of dry fractionation briefly.
First paragraph can be improved by explaining the current situation of obesity using more references from the recent studies and statistics.
Some recent studies and their results can be briefly given in fourth paragraph. The references given are from 1985, 1995, 1999. What about the recent publications?
In last paragraph, the authors should explain why they conducted the study, what is unique and novel in this study prior to what they did.

Materials: Well-prepared. Small changes is needed.
What is Novozyme 435? Is it an enzyme/ lipase? Should be given.
Units should be consistent through the text. Use just min or minutes, hours or h, mL or ml?
Analyses are conducted duplicates, triplicates? Should be given.

Results:
Results part should be improved.

In this part when the authors evaluate the effects of parameters on the chemical properties, they should give the most important and significant results first by using basic sentences. They should not start to explain using the numbers and complex, long sentences. Then, they should explain the probable reasons in details using the recent publications as references.

Tables:
Table 1 cannot be understood easily. What the varying Ct, Cr and varying as mean? You can briefly simplify/summarize each experimental set using the variables and constants. For example, for the parameter Ct you can put the variables (Ct) (30-50) and constants (using 35C/min cooling rate and … rpm).

In Table2: it is difficult to see the conditions evaluated. First column should be the layers, two rows including olein and stearin. Second column should be crystallization temp and you should put 30, 35, 40, 45, 50C for each layer (olein and stearin) and remove sample column.
Correct the third and fourth tables as described above. For third and fourth one, add a column for cooling rate and agitation speed, respectively.
Describe the abbreviations SAFA, MUFA, PUFA as footnotes besides Ol and St.
Explain what are a, b, c in Table 2, 3 and 4 as footnote.
The results can be given using one decimal after comma instead of two decimal to simplify.

Figures:
Figure 1, 2 and 3 should be given under Figure 1 as 1A, B and C to see and evaluate the effects of parameters and compare their effects easily. Same should be done figure 4, 5 and 6 and can be given under Figure 2 as they all related about DSC thermo profile.
Figure 4: 4;1.00C/min

Conclusion:
It is written as a summary of results. It should be rewritten.

Some vocabularies & grammar correction should be done through the text:

Pg7 line 17 : as expected instead of as can be expected. were present instead of are present.
Pg7 line 24: reduction instead of significant reduction
Pg 8 line 4: Remove it is supported by the fatty acid composition shown in Table 2. Just put table2 in brachets at the end of the sequence in line 4.
Pg 8 line 6: a comma should be added after C18:1
Pg 8 line 10: instead of this is influenced ….. it might be as “this might be resulted from the increasing palmitic acid concentration. This demonstrate that olein…..
Pg 8 line 13: instead of (Arnaud & Collignan, 2008) use Arnaud & Collignan (2008)
Pg 8 line 15: cut at low temperatures and put at the end of the same sentence.
Pg 8 line 15: Sequence can be simplified as: In a study of Mamat et al. (2005) on palm and sunflower oil blends fractionation using different temperatures, it was also reported that the higher IV can be obtained due to higher PUFA proportion found in liquid fraction when lower fractionation temperature was applied.
Pg 8 line 21: in this study refers to Mamat et al (2005) or the present study? If it is about yours use the present study instead of this.
Pg 8 line 22: this is probably related to an inconsistency in separation processes.
Pg 8 line 23: instead of time use duration
Pg 9 line 1: an increase in olein entrapment might have contributed to an increase in IV of stearin…
Pg 9 line 7: compositions
Pg 9 line 19: Remove it is best explained… better to say “The increment in IV…..
Pg 9 line 22: No need to say under this conditions.
Pg 10 line 22: palmitic acid, respectively, i.e. higher MUFA and lower SAFA contents were detected.
Pg 10 line 2: can be explained better.
Pg 10 line 6: can be written better
Pg 11 line 3: Remove brachet before the author’s name and put the year in a brachet.
Pg 11 line 7: The stearin yield is strongly dependent on the …… (Table 2, 3 & 4).
Pg 11 line 8: No need to say “one can observe that”
Pg 11 line 12: Herrera & Hartel (2000)
Pg 11 line 15 Authors (year)
Pg 11 line 18: The acylglycerol composition can be altered from the feed oil as expected. The proportion of TAG was higher in the olein,……. (Table 2, 3 & 4).
Pg 11 line 21: Remove “rather small” use insignificant
Pg 11 line 21: remove “and” use , hence it was not investigated.
Pg12: Paragraphs shoulb be collected in one paragraph.
Pg 12 line 1: Cooling rates
Pg 12 line 10: Remove first sentence. Put (Figure 2) after the second sentence.
Pg 12 line 15: Remove first sentence. Put (Figure 2) after the second sentence.
Pg 12 line 16: do not need to put it was found that
Pg 13 line 2: comma after PDAG
Pg 13 line 5: This is due to an increase in the amount ….
Pg 13 line 19: an increase
Pg 14 line 2: does not influence the SMP of the stearin. An increase in….

Experimental design

Experimental design is good but the presentation of the results should be improved and discussed further. Methods used are explained well.

Validity of the findings

Results are robust, statistically sound and controlled. The conclusion part should include the answers of original question investigated, it should be rewritten.

---

## Round 0.2 · accepted · Accept

The reviewers' comments have been well addressed.